# Sociological variety and the transmission efficiency of *Mycobacterium tuberculosis*: a secondary analysis of qualitative and quantitative data from 15 communities in Zambia

Emma J Murray ,[1] Peter J Dodd,[2] Ben Marais,[3] Helen Ayles,[4,5] Kwame Shanaube,[5] Albertus Schaap,[6] Richard G White,[7] Virginia Bond[5,8]

For numbered affiliations see end of article.

**Correspondence to**
Emma J Murray;
emmajane.murray@gmail.com

## ABSTRACT

**Objectives** Selected Zambian communities formed part of a cluster randomised trial: the Zambia and South Africa TB and AIDS Reduction study (ZAMSTAR). There was wide variability in the prevalence of *Mycobacterium tuberculosis* infection and tuberculosis (TB) disease across these communities. We sought to clarify whether specific communities could have been more/less vulnerable to *M. tuberculosis* transmission as a result of sociological variety relevant to transmission efficiency.

**Design** We conducted a mixed methods secondary analysis using existing data sets. First, we analysed qualitative data to categorise and synthesise patterns of socio-spatial engagement across communities. Second, we compared emergent sociological variables with a measure of transmission efficiency: the ratio of the annual risk of infection to TB prevalence.

**Setting** ZAMSTAR communities in urban and peri-urban Zambia, spanning five provinces.

**Participants** Fifteen communities, each served by a health facility offering TB treatment to a population of at least 25 000. TB notification rates were at least 400 per 100 000 per annum and HIV seroprevalence was estimated to be high.

**Results** Crowding, movement, livelihoods and participation in recreational activity differed across communities. Based on 12 socio-spatial indicators, communities were qualitatively classified as more/less spatially crowded and as more/less socially 'open' to contact with others, with implications for the presumptive risk of *M. tuberculosis* transmission. For example, watching video shows in poorly ventilated structures posed a presumptive risk in more socially open communities, while outdoor farming and/or fishing were particularly widespread in communities with lower transmission measures.

**Conclusions** A dual dynamic of 'social permeability' and crowding appeared relevant to disparities in *M. tuberculosis* transmission efficiency. To reduce transmission, certain socio-spatial aspects could be adjusted (eg, increasing ventilation on transport), while more structural aspects are less malleable (eg, reliance on public transport). We recommend integrating community

## Strengths and limitations of this study

► We employed a novel qualitative approach to investigate disparities in transmission efficiency.
► We combined population (community) level socio-spatial data with actual measurement of tuberculosis (TB) infection and disease.
► Data were effectively cross-sectional, focusing on interactions within localised geographical boundaries, and did not capture details of extra-community travel, nor of infrequent and/or seasonal events, potentially relevant to TB transmission.
► Results are context specific; therefore, further investigations in other settings would be valuable to assess the generalisability of key concepts.
► Measures of association should be interpreted with appropriate caution and the study regarded as hypothesis generating.

level typologies with genome sequencing techniques to further explore the significance of 'social permeability'.
**Trial registration number** ISRCTN36729271.

## INTRODUCTION

Interactions between the tuberculosis (TB) and HIV epidemics have complicated and compromised TB containment efforts in Zambia and other high burden countries. There is evidence that as the HIV epidemic ages, an increasing burden of undiagnosed TB is emerging among older adults in Zambia, underscoring the resilience of epidemic TB in the region.[1–5] Understanding the role of socioeconomic factors in facilitating the transmission of *Mycobacterium tuberculosis*, within and outside households, is critical to developing alternative approaches that could reduce ongoing *M. tuberculosis* transmission at the community level.[6–11] This inquiry explores how socio-spatial aspects of

everyday community life might be relevant to differing *M. tuberculosis* transmission dynamics. We used a novel mixed methods interdisciplinary approach to conduct a secondary analysis of Zambian data sets, collected during a cluster randomised trial (CRT)—the Zambia and South Africa TB and AIDS Reduction study (ZAMSTAR)—and as part of an ancillary study, named the Contact Observations of Daily Activities (CODA).[12 13] The approach combined community level qualitative data, focused on the use of space and daily social interactions, with quantitative measurements of TB prevalence and annual risk of infection (ARI), to investigate whether differences in socio-spatial aspects across communities corresponded to a higher or lower *M. tuberculosis* transmission efficiency (measured as the ARI per prevalent TB case in a community).

ZAMSTAR (2004–2011) compared two interventions to reduce *M. tuberculosis* transmission and TB prevalence across Zambia and South Africa at a population level. The two interventions—enhanced community TB case finding and household level combined TB–HIV care—were implemented from 2006 to 2009. In Zambia, 16 communities were selected by TB notification rates in excess of 400 per 100 000 per annum and a high HIV prevalence.[12 14–16] Baseline prevalence of *M. tuberculosis* infection, as measured by tuberculin skin test (TST) surveys in primary school children, was highly variable, ranging from 2% to 13% across the Zambian communities and allowed for the estimation of ARI.[14] The final ZAMSTAR prevalence survey found large variation in culture-confirmed TB prevalence, from 221 to 1095 per 100 000 in the Zambian communities. This variability implied, in part, differences in *M. tuberculosis* transmission dynamics. Towards the end of ZAMSTAR, CODA sought to measure patterns of social contacts relevant to *M. tuberculosis* transmission and model the implied ARIs in adult age groups.[13] Our inquiry supplements the CODA quantitative analysis of adult social contacts, investigating social and spatial contextual variables that may influence community level *M. tuberculosis* transmission.

## METHODS

This secondary analysis used whole communities, rather than the individual, as a unit for analysis and drew on four sources of data: qualitative data collected for the purpose of the CODA study; estimates of population density from the ZAMSTAR household enumeration; estimates of ARI from ZAMSTAR TST surveys; and TB prevalence data from the final ZAMSTAR TB prevalence survey. Details of ZAMSTAR household enumeration, TST data and implied ARI estimates, TB prevalence data and linked methods have been published.[12 14 16] The Zambian communities investigated in this secondary analysis were preselected as a set of 16 for the ZAMSTAR trial. The ZAMSTAR research design needed the potential to detect statistically significant reductions in TB prevalence and infection incidence. Therefore, communities were

selected based on prespecified criteria, including TB case notification >400 per 100 000 per annum. A further prerequisite was that TB diagnostic and treatment service were broadly equivalent (de facto functioning local health facilities provided by the state, with trained healthcare providers and voluntary TB supporters, serving a population of at least 25 000). Given the role of HIV infection in driving TB cases in this setting, ZAMSTAR additionally selected on the basis of relatively high burdens of HIV (compared with the national average). In the absence of population level data on HIV, proxy information was used to identify communities, drawing on the opinion of local experts, other survey sources and seeking the buy-in and advice from the Ministry of Health. The resulting Zambian ZAMSTAR communities were distributed across 5 provinces and districts with subsequent measures by the ZAMSTAR trial showing that HIV prevalence ranged between 8.1% and 26.6%.[12] All communities were predominantly high-density urban communities (three were peri-urban) located along the line of rail and/or major trading routes. All communities had a broad mix of ethnic groups and four communities had a stronger, yet insignificant presence of non-nationals due to the proximity of international borders and location within the capital city. Socioeconomic status was mainly classified as low, with some pockets of lower middle class in all communities.

We focus on the CODA qualitative data collection and analysis used for the identification of socio-spatial variables; our use of quantitative ARI and prevalence data from ZAMSTAR; and the mixed method analysis approach that investigated relationships between socio-spatial variables and calculated *M. tuberculosis* 'transmission efficiency'.

### Qualitative data

Following the intervention period of ZAMSTAR, Broad Brush Surveys (BBS) were conducted in 2011 across Zambian ZAMSTAR communities as part of CODA.[17] BBS is a method which lends itself to the systematic sociological comparison of bounded urban places (in this instance communities) across socio-space. The development of meta-indicators around a core research question allows for abstraction and a degree of generalisability.[17–20] The CODA BBS aimed to (a) update previous BBS data collected as part of ZAMSTAR between 2004 and 2005[19] and (b) provide insight into patterns of behaviour and settings of adult-to-child social mixing, in order to inform the design of a quantitative questionnaire.[13] The BBS fieldwork was completed by 32 research assistants trained over a 2-day period in participatory methods. Fieldwork was conducted by a team of 2 (usually 1 man and 1 woman) over a period of 5 days in each community, using local languages. In sequence, each pair carried out a transect walk through the hub of the community; structured observations of different gender and age spaces (older men/women, younger men/women and boys/girls); a structured observation of the health facility,

church services and the main transport depot; and day long household observations (two per community). The focus of these observations were the presence and contact between adults and children. Daily activity charts were drawn up with boys and girls during the observations, in their spaces. Each team documented observations and conversations in the form of a written 'activity report'. Activity reports were semi-structured, open-ended and written in English, with translation completed in the process of writing by the relevant fieldworkers. Research teams were debriefed on a daily basis, both within the pair of research assistants and remotely over the phone. The second and last authors were part of the training and fieldwork.

The qualitative BBS data indicated how public places and transport were used by different age groups and genders within each community. Our focus on the broader community is justified as we understand transmission is less likely to take place within households in this particular setting.[21] For the purposes of this analysis, we identified and categorised patterns of social and spatial engagement using conversational and observational material from the BBS activity reports, until a point of saturation was reached for each community. This was done in frequent consultations with a key member of the fieldwork team, who also guided the first author in conducting a relational comparison of the patterns identified across communities. One of the 16 Zambian ZAMSTAR communities was dropped from our analysis due to poor data quality. Similarities and differences were subsequently compared across the remaining 15 communities. We configured findings into an annotated tabular format of socio-spatial tendencies, broken down across age and gender categories where appropriate (table 1).

We delineated the strongest variables as broadly linked to one or more of the following aspects: (a) levels of spatial crowding, pertaining to the amount of people within shared/confined spaces and the density of physical structures; (b) the range and degree of participation in predominantly indoor recreational activity; (c) the dominance of outdoor livelihoods, for example, widespread commitment to farming or fishing dramatically shaped socio-spatial engagement in certain communities; and (d) what we have termed as 'social permeability', that is, the level of exchange between, or movement of, local residents and non-residents across and within geographical boundaries, with some communities clearly more socially 'open' to extra-community contact. Eleven qualitative indicators were identified to represent assessments of socio-spatial variability, selected on the basis of being observable (tangible) and categorical, with perceived relevance for TB transmission. The indicators were general crowding in public spaces/places; the use of overloaded and poorly ventilated transport; local residents remaining in their residential community during the day; housing density; crowding at health centres; time spent at home during the day; watching video shows; going to bars/taverns; spending time farming/ fishing; the presence of non-residents; and evidence of residents travelling out of their residential community.

## Quantitative analysis

As part of ZAMSTAR, TST surveys were conducted in a cohort of primary school children to calculate ARI across communities, estimating community ARIs of between 0.25%/year and 1.49%/year (using a 15-millimetre cut-off for positivity).[16] For our mixed method analysis, we calculated the ARI-to-TB prevalence ratio, which is an established measure of 'transmission efficiency' (beta, in infectious disease modelling) and likely to be influenced by social features that affect the patterns and intensity of mixing relevant to *M. tuberculosis* transmission. Karel Styblo used this ratio to help describe historical patterns in European TB epidemiology. While the values that Karel Styblo found no longer apply, the ratio remains a useful metric.[22] We calculated this ratio for each community using calculated ARI point estimates and TB prevalence point estimates determined during ZAMSTAR. We also computed Kendall Rank correlation coefficients (Spearman correlation coefficient for the quantitative indicator, population density) and approximate p values. Given the number of variables considered, it is not appropriate to consider p values below a certain threshold as 'significant'; they simply provide some guidance on how likely it is an apparent individual pattern that could arise by chance.

## Mixed methods analysis

After completing the qualitative analysis and quantitative calculations described above, we first correlated the measure of transmission efficiency with the 11 qualitative variables. Each variable was titled according to one of the four social aspects to which it most strongly indicated. For example, the variable 'presence of non-residents' was regarded as a strong indicator of social permeability, given the manner in which it presented in the BBS. Qualitative variables (indicators) were also ranked by a priori expectations of increasing transmission risk. For example, 'high'>'low' for housing density, or 'WMGB'>'M' for women, men, girls and boys spending time in bars versus only men. Previous work on these communities combining the TB prevalence results, ARI estimates and quantitative social contact data suggested men are responsible for more infections than women.[13] Children, especially aged <5 years, typically have lower bacteriological load than adults and are likely to contribute less to transmission than adults of either sex. We also included an objective quantitative indicator of crowding, population density, measured as the number of people per square kilometre (based on the ZAMSTAR household enumeration). To better understand the relationships between community indicators and the ARI-to-prevalence ratio, we constructed corresponding graphical outputs (online supplemental material) for ARI (online supplemental figure S1) and prevalence separately (online supplemental figure S2).

**Table 1** Relational categorisation of socio-spatial tendencies across communities

| Community ID number | 1 | 2 | 3 | 4 | 5 | 6 | 7 | 8 | 9 | 10 | 11 | 13 | 14 | 15 | 16 |
|---|---|---|---|---|---|---|---|---|---|---|---|---|---|---|---|
| **CROWDING** | | | | | | | | | | | | | | | |
| General crowding in public spaces/places | Yes | Yes | Yes | Yes | Yes | Yes | Yes | Yes | Mixed | Yes | Yes | No | Mixed | No | No |
| Use of overloaded and poorly ventilated transport | Yes | Uncertain | Yes | Yes | Yes | Yes | Yes | Yes | Yes | No | Yes | Yes | No | No | No |
| Locals remaining in community (daytime) | Negligible | Negligible | Negligible | M W B G | Negligible | W G B | M W G B | M W G B | Negligible | Negligible | W | B G | W G B | W G B | B G |
| Housing density | Mixed | Low | High | Mixed | High | High | Mixed | High | High | Mixed | Mixed | Low | Low | Low | Low |
| Crowding at health centre | Yes | No | Yes | Yes | Yes | Yes | Yes | Yes | No | No | Yes | No | Yes | No | No |
| Crowding at churches | Yes | Yes | No | Yes | Yes | Yes | Yes | Yes | Yes | Yes | No | Yes | No | No | No |
| **INDOOR ACTIVITY** | | | | | | | | | | | | | | | |
| At household (daytime) | M W B | W B | W B | W | W | W G | M W B G | B G | W | W | M W G | W B G | M W B G | M W G | W B G |
| Watching video shows | M G B | B | B | M G B | M W B G | M W B G | M W B G | M W B | W M | G B | M W B G | B | Negligible | M | G B |
| Time at bars/taverns | M W B G | M | M | M W B G | M W B G | M W B G | M G B | M W | M W | M W B G | M W | M W | M W B G | Negligible | M |
| To church | M W B G | M W B G | M W B G | M W B G | M W B G | M W B G | M W B G | M W B G | M W B G | M W B G | M W B G | M W B G | M W B G | M W B G | M W B G |
| **ECONOMY** | | | | | | | | | | | | | | | |
| Residents farming/fishing | Negligible | W G B | W M B | Negligible | M | W | Negligible | W M B | W G B | M W B G | M W B G | W M | M W B G | M W G | Negligible |
| Residents at local market | M W | M | Negligible | M G B | W B | M G B | W G B | M W | W | W | W G | W | Negligible | M W | W |
| Residents using local shops/stalls | M | Negligible | M W | M G B | M | M W | M | M | Negligible | M | Negligible | M | M | M | M |
| **SOCIAL PERMEABILITY** | | | | | | | | | | | | | | | |
| Presence of non-residents | M W | W | M W B G | M W B G | M | M W | M W B G | M W | Negligible | Negligible | Negligible | Negligible | W M B | Negligible | Negligible |
| Residents travelling out of community | M W B G | M W B G | M W | M W B G | M W B G | M W B G | M W B G | M W | M W | M W B G | M | M | M | M W | M |
| Residents travelling to external market | M W | Negligible | W | M W | W | M W | M W | Negligible | W | W | Negligible | W | Negligible | W | W |
| Social mixing (general) across age and gender | Some | Yes | Yes | Yes | Yes | Some | Some | Yes | Some | Yes | Least | Some | Yes | Less | Least |
| At health centre | M W B G | Negligible | M W B G | W G B | M W B G | M W B G | W G B | M W B G | W | W | M W B G | W G B | W G B | W | W B G |
| **OTHER** | | | | | | | | | | | | | | | |
| Children not attending school | Negligible | Negligible | Negligible | B G | B G | Negligible | B G | Negligible | Negligible | Negligible | Negligible | Negligible | Negligible | Negligible | B G |
| To salon/barber | W | Negligible | Negligible | W | Negligible | M W G | W | Negligible | W | W | W | Negligible | W | W | W |
| To outdoor water point | W G | Negligible | Negligible | W G | W | Negligible | W G B | W | Negligible | W G | Negligible | G | Negligible | Negligible | W |
| To outdoor recreation | B | B | M | M B G | M B | Negligible | M B G | B | Negligible | B | B G | B | M B | M B | B G |

Rows of variables were identified and thematically grouped following qualitative BBS data analysis. Themes are capitalised.
Each column provides a 'snapshot' of general tendencies within a particular community. Qualitative assessments of variables were relationally gauged across communities, using BBS data.
B, Boys; BBS, Broad Brush Surveys; G, Girls; M, Men; W, Women.

## Patient and public involvement

Community Advisory Boards (CABs) were consulted throughout the ZAMSTAR trial and formed before the implementation of interventions. In the 16 Zambian communities, CABs were formed from pre-existing Neighbourhood Health Committees (NHCs) in each community. NHCs were already representative of different geographical 'zones' in the trial communities, endorsed by an Act of Parliament and democratically elected. NHCs were, therefore, asked if they would like to represent public (including patient) TB and HIV interests, and sought representation across interest groups, age and gender. Although not involved in the design of the ZAMSTAR trial, CABs were closely involved from ZAMSTAR baseline research onward. Zambian research work was conducted by Zambart, a research organisation housed by the University of Zambia's School of Medicine and with links to the London School of Hygiene and Tropical Medicine. CAB members received training in good clinical practice, research protocols and research dissemination. They were involved in the public randomisation of the trial communities and all research dissemination. Other than the CABs, representative participants involved in different components of the trial were invited to community and district dissemination events. This included BBS disseminations, carried out in the communities, using informative flyers for community members. Being a substudy of ZAMSTAR, CODA used the same CAB structures and communication channels. The community engagement lead, Musonda Simwinga, used his experience with the ZAMSTAR CABs as a platform for his own PhD on community engagement using CABs in trials, including reflections on equitable partnerships and accountability, and to develop other CAB structures for subsequent trials.[23]

## RESULTS

Table 1 shows the relational categorisation of socio-spatial findings from the BBS CODA data, that is, it is a structured representation of the qualitative analysis. The qualitative findings are described below. Figure 1 merges socio-spatial variables with the calculation of transmission efficiency across study communities. It shows the relationship between the ARI-to-prevalence ratio (y axis) and 12 indicators (11 qualitative and 1 quantitative). Similarly, online supplemental figures S1 and S2 separately show the same indicators in relation to ARI and TB prevalence, respectively. Indicators (variables) were typically more tightly linked with ARI, and less tightly linked with disease prevalence, than they were with the ARI-to-prevalence ratio. The stronger link between the indicators and ARI may reflect that ARI is a cumulative measure, based on relatively large numbers, whereas disease prevalence is a cross-sectional measure, with relatively small numbers of cases in each community that may fluctuate randomly over time.

## Variations in community life

Certain communities were classified as more 'socially permeable' than others: socially 'open' to exchange across geographical boundaries and orientated toward mixing across age and gender in congregate settings. In nine communities, adults from other regions—'outsiders'— engaged more with local community life. The presence of 'outsiders' appeared particularly diverse in three of these nine communities, where men, women, girls and boys visited. In contrast, there was little to no evidence of non-residents operating in the remaining six communities. In places with a strong presence of 'outsiders', social mixing across age groups and genders appeared widespread, although where and how this contact took place varied. For example, in community 14, bars played a greater role in bringing people together than video shows. We also distinguished between communities where both adults and children travelled out and those where mostly adults travelled out. In four communities, only men travelled out regularly. This travel could be linked to livelihood pursuits, often of an agricultural nature. Involvement in seasonal farming or fishing was particularly widespread across age and gender in four communities; in an additional seven communities, there was evidence of engagement with outdoor farming or fishing, but this was notably less intense. Residents left communities for short periods to fish and/or buy fish or tend to fields in the rainy season. There was no evidence of farming or fishing in the remaining 4 (of the 15) communities.

Levels of crowding varied at health centres and in other gathering places, although the majority of communities were classified as generally 'crowded' by field-workers. The same was true for housing density: only five communities were observed to have relatively low-density housing. Importantly, public transport was often overloaded and gauged as crowded, with extended durations of travel. Ventilation during travel was ordinarily poor given that it was common practice to keep windows shut to prevent dust. In one community, panel vans with no windows were being used. Although most communities were reliant on buses, minibuses and/or vans for travel, some were substantially less dependent on public transport; residents of one community mostly travelled by foot. When assessing who remained within the geographical boundaries of their residential community during the day, communities again differed. In six communities, we found little evidence of residents remaining. Children and/or women generally remained in six of the other nine communities, but men rarely did. In three communities, local men were significantly more present.

Proportionally, women generally spent more time at home (in all communities except one). In some communities, men appeared more engaged with recreational activity. Indoor recreational activities across communities were often similar, but contact patterns differed. Children came into contact with adults at drinking places and/or at pool table clubs in seven of the communities, while in the others this was far less evident. In three communities,

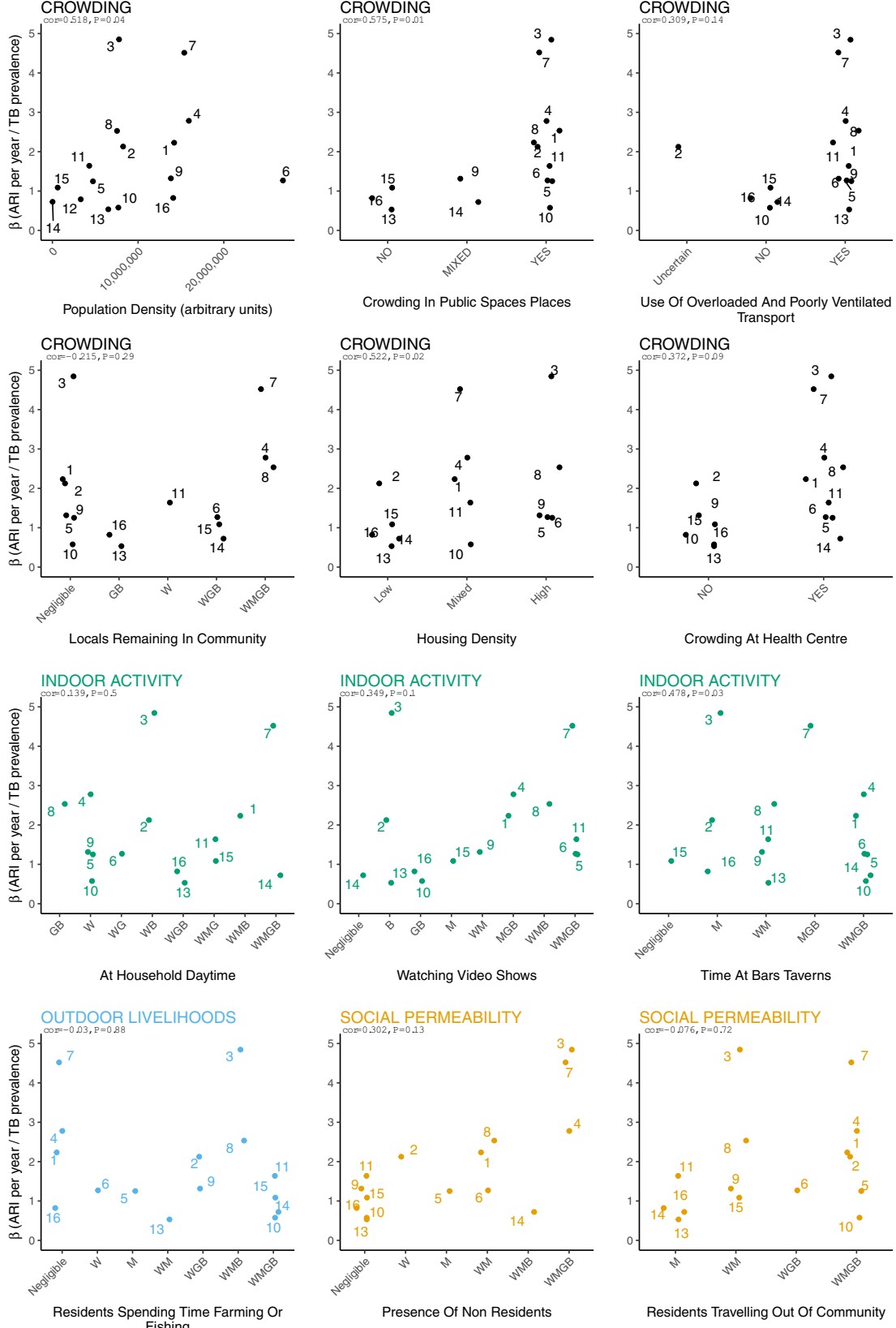

**Figure 1** Transmission efficiency (ratio of annual risk of infection to disease prevalence) by socio-spatial variables.

only men were ever seen at bars. In certain communities, women were observed with babies strapped to their backs and boys were frequently trading in eggs and/or bottles, highlighting the multifunctionality of drinking places in

these areas. Bars and taverns were generally crowded and poorly ventilated, with limited small windows designed to deter theft. Watching films in small makeshift tented 'theatres' was a popular leisure activity in 14 communities.

Shows were held in small outdoor structures, erected using poles and tarpaulin, measuring approximately 4×2.5 m. These structures were darkened as much as possible to enhance viewing, by closing gaps in the tarpaulin, which in turn limited ventilation. Audiences were ordinarily seated on a few tightly packed benches. Children watched video shows during the day, and in the evening, adults and children often watched together. All communities showed widespread commitment to their local churches. Community members of all ages spent a substantial proportion of their time at services ranging between 2 and 6 hours, which often exceeded the time spent on many of the other contact activities identified. It should be noted that most community residents are Christian and the fieldwork was conducted in the period leading up to Easter.

### Patterns with respect to transmission efficiency

Population density and qualitative assessments of crowding were broadly concordant, given the small sample size (number of communities). Communities sharing higher transmission efficiency could generally be classified as more crowded, whereas communities with lower transmission efficiency tended to present in more of a mixed way (across lower and higher degrees of crowding). By tracing an individual community (number) through figure 1, one is able to consider the interaction of the different socio-spatial variables identified. Where transmission efficiency was highest, there were also higher levels of social permeability. This is illustrated by four communities (3, 4, 7 and 8), where a wide range of non-residents were present and resident adults frequently travelled out. Moreover, communities 4, 7 and 8 showed evidence of social mixing across age and gender in local recreational activity as well as having a number of residents who remained in the community during the day, suggesting a diverse range of contact opportunities. In terms of social mixing, community 3 was distinctive. While 'outsiders' came in and locals travelled out, there was a significant pattern of extra-community travel and many locals did not remain in the community during the day. Furthermore, social mixing during recreational activity appeared relatively limited. This reflects particular characteristics of the community. The community had a large local market, attracting 'outsiders', and a large proportion of established lower middle-class residents (including police and army) who travelled out to work, creating a substantial daily population 'swap'. Class distribution also undermined diversity in local social mixing (with middle class having a preference for socialising outside of their community). Overall, the social permeability of these four higher efficiency communities contrasted with the lower transmission efficiency seen in the most socially closed communities, with far fewer 'outsiders' and less extra-community travel. Widespread participation in farming and/or fishing (across age and gender) seemed to correspond with only lower levels of transmission efficiency, while partial or no time spent farming/fishing corresponded with both higher and lower transmission efficiency. There was no pattern across indicators of wider participation in indoor activities; however, where only children watched video shows, transmission efficiency appeared lower, with clear exceptions of communities 2 and 3.

## DISCUSSION

This interdisciplinary analysis suggests that certain communities may have been protected from *M. tuberculosis* transmission in unique and sociologically complex ways. Communities with lower measures of transmission efficiency included, but were not limited to those in which fishing and/or farming were particularly dominant across gender and age; where public transport was more ventilated and less overloaded; where less crowding took place in public space; where fewer 'outsiders' were present; where only men mostly travelled out of their community; where there was negligible evidence of indoor recreational activity; and where population and housing density was lower. Links between some of these socio-spatial aspects and *M. tuberculosis* transmission risk have been shown, using different approaches, in other sub-Saharan settings. In Malawi, Odone *et al* identified that TB was more common among those working in the cash economy than those in subsistence economy,[24] and studies in South Africa have highlighted the significance of confined public transport and gathering places, with emphasis on the implications for air hygiene and rebreathed air volumes.[25–27]

From a distinctly qualitative angle, our work begins to unveil and define how the sociological topography of a particular community could play an important role in shaping transmission efficiency. Two meta-observations are significant in this regard. The first observation relates to the relative vulnerability of the communities with more outwardly orientated and socioeconomically diverse features in comparison to their more socially secluded and/or agriculturally focused counterparts. Certain communities exhibited a dual dynamic of crowding and 'social permeability', which could be providing increased opportunity for the importation of *M. tuberculosis* and local aerosol transmission in poorly ventilated spaces. 'Social permeability' not only reflects the presence of non-residents within a community, but it also captures the reach of 'mixing' across age and gender, and patterns of locals travelling out. The latter provides additional opportunity for extra-community *M. tuberculosis* exposure and infection (or re-infection), outside of the residential community. It seems that this dual dynamic of spatial 'closeness' and social 'openness' is relevant

to increased *M. tuberculosis* transmission across the communities.

The second observation relates to a distinction that needs to be drawn between what we could call 'structural' and 'routine' variables. By structural variables, we refer to socioeconomic factors that are to a certain extent beyond people's agency. For instance, communities that are more structurally dependent on exchange and interaction with outsiders could be more vulnerable to *M. tuberculosis* exposure. Routine variables refer to habits or patterns of behaviour, which, although related to socioeconomic factors, are not obviously reducible to them. Watching video shows is a good example; it is a form of routine leisure activity which carries a particularly high presumptive risk for *M. tuberculosis* transmission because of the darkened, poorly ventilated and contained space in which it is conducted. For health policy, our distinction between 'structural' and 'routine' variables is important because the latter would be more malleable to intervention. While structural dependency on the outside world cannot be altered easily, people can be encouraged to change the way in which they conduct routine leisure activities; for instance, a concerted effort could be made to increase ventilation in these places.[28]

Our secondary analysis has limitations. Figure 1 is exploratory and hypothesis generating, and measures of association must be interpreted with appropriate caution. Nonetheless, results do suggest tentative patterns with respect to some indicators worthy of attention in future work. The estimates of ARI and TB prevalence are of limited precision, and ARI estimates can depend strongly on the method of TST interpretation. Empirical estimates[29] and modelling[13] suggest ARI estimates in children are lower than in adults. However, ARI-to-prevalence ratios should still hold validity for relative comparisons between communities, since age-related contact patterns did not vary substantially between the communities.[13] We emphasise that our analysis did not include all community features relevant to socio-spatial engagement. Congregation linked to both schools and churches has been established as a transmission risk factor in similar high burden settings,[30 31] yet we were unable to conduct BBS at schools. Importantly, churches emerged as significant congregate features, with widespread attendance presenting in similar ways across age, gender and time. This consistency, across all communities, meant that churches could not be linked to sociological variation in a meaningful way. We are also aware that local mines and prisons may have an important impact on *M. tuberculosis* transmission risk in communities[32–36]; however, we were unable to identify any significance in our data. Additionally, there was scant evidence of women or men regularly attending traditional ceremonies, which may require further consideration. The observational BBS did not hold enough ethnographic

depth to capture the pervasiveness of guarded socio-spatial activities.

Findings are context specific and data were effectively cross-sectional. We focused on interactions within localised geographical boundaries, and did not capture interactions on the far side of extra-community travel. Furthermore, fieldwork was conducted at a particular time of year and observations took place during the daytime, which limits the generalisability of our observations accordingly—not only across but also within the study communities. We anticipate relevant socio-spatial features and their role in mediating TB transmission will vary between contexts. However, the dual dynamic of spatial closeness (crowding) and social openness (permeability) may be relevant in other settings, with further investigations valuable. The addition of pathogen genomic data might help generate or eliminate hypotheses on transmission location or context, and recent research illustrates that quantitative measures of geographic mobility could be applied to studies of social permeability.[37] Whole genome sequencing may clarify whether importation is more or less relevant in certain communities and highlight transmission chains.

## CONCLUSION

Localised sociological distinctions offered an explanation of why certain communities might have been more (or less) vulnerable to *M. tuberculosis* transmission. Interactions between social permeability and crowding appear relevant to increased transmission efficiency, at least in these Zambian communities. A conceptual framework differentiating structural and routine elements of high-risk socio-spatial features could provide a valuable foundation for the development of contextualised public health interventions. For example, some socio-spatial aspects carrying transmission risk are easily adjusted (eg, increasing ventilation on transport or at video shows), while more structural aspects are less malleable (eg, reliance on public transport). The significance of social permeability could be further explored by integrating community level typologies with genome sequencing techniques able to map strain introduction events and community transmission chains.

**Author affiliations**
[1]Independent researcher, Cambridge, UK
[2]School of Health and Related Research, The University of Sheffield, Sheffield, UK
[3]Children's Hospital Westmead Clinical School, The University of Sydney, Westmead, New South Wales, Australia
[4]Clinical Research Department, Faculty of Infectious and Tropical Diseases, London School of Hygiene and Tropical Medicine, London, UK
[5]Zambart, School of Public Health, University of Zambia, Lusaka, Zambia
[6]Department of Infectious Disease Epidemiology, London School of Hygiene and Tropical Medicine, London, UK
[7]TB Modelling Group, Department of Infectious Disease Epidemiology, London School of Hygiene and Tropical Medicine, London, UK

[8]Department of Global Health and Development, Faculty of Public Health and Policy, London School of Hygiene and Tropical Medicine, London, UK

**Acknowledgements** We extend thanks to the communities, all participants, the regulatory authorities, the Ministry of Health and Zambart staff in Zambia for their support and involvement via the CODA and ZAMSTAR studies. In particular, Levy Chilikwela provided invaluable assistance with qualitative data analysis. We are grateful to Patrick Baert, Peter Godfrey-Faussett and Jean Hunleth for feedback on various versions of this paper. Additionally, comments and recommendations from the journal's reviewers during the peer review process helped sharpen our report. Emma Murray would like to thank the Desmond Tutu TB Centre, Department of Paediatrics and Child Health, Stellenbosch University, South Africa, for all manner of things. Finally, we would like to thank the principal investigators of ZAMSTAR who are not co-authors, namely: Nulda Beyers, Peter Godfrey-Faussett and Richard Chaisson.

**Contributors** Substantial contribution to conception and design of Zambia and South Africa TB and AIDS Reduction study: HA, VB and KS. Substantial contribution to conception and design of Contact Observations of Daily Activities study: HA, VB, PJD and RGW. Substantial contributions to conception and design of secondary analysis: VB, PJD, BM, EM and RGW. Acquisition of study data: HA, VB, PJD, AS and KS. Analysis and interpretation of data: PJD and EM. Drafting of article: EM. Revising critically for intellectual content: HA, VB, PJD, BM, EM and RGW. Guarantor: VB.

**Funding** This secondary analysis received no specific grant from any funding agency in the public, commercial or not-for-profit sectors. However, HA, VB, PJD, EM and RGW were funded by the Bill and Melinda Gates Foundation (Consortium to Respond Effectively to the AIDS/TB Epidemic: 19790.01). RGW is funded by the Wellcome Trust (218261/Z/19/Z), NIH (1R01AI147321-01), EDTCP (RIA208D-2505B), UK MRC (CCF17-7779 via SET Bloomsbury), ESRC (ES/P008011/1), BMGF (OPP1084276, OPP1135288 and INV-001754) and the WHO (2020/985800-0). PJD is supported by the UK Medical Research Council (MR/P022081/1). (This UK-funded award is part of the EDCTP2 programme supported by the European Union.)

**Disclaimer** Funders had no involvement in the design, collection, analysis or interpretation of the data, in writing the manuscript or in the decision to submit.

**Competing interests** None declared.

**Patient consent for publication** Not applicable.

**Ethics approval** Formal ethical approval was not required for this secondary analysis of existing data. Earlier approval for fieldwork was obtained through the Contact Observations of Daily Activities study from the University of Zambia's Biomedical Research Ethics Committee (007-10-04), and London School of Hygiene and Tropical Medicine's (LSHTM) Ethics Committee (A211 3008). The Zambia and South Africa TB and AIDS Reduction study was approved by ethical review boards of the University of Zambia, Stellenbosch University and LSHTM.

**Provenance and peer review** Not commissioned; externally peer reviewed.

**Data availability statement** Data are available upon reasonable request. Qualitative BBS data are available upon reasonable request from Virginia Bond (ORCID: 0000-0002-6815-4239)Quantitative ZAMSTAR data are available upon reasonable request from Albertus Schaap (ORCID: 0000-0002-5959-526X)Extra CODA study data are at Zenodo. DOI: http://doi.org/10.5281/zenodo.3874675. Relevant license details are there and have Creative Commons Attribution (https://creativecommons.org/licenses/by/4.0/legalcode).

**ORCID iD**
Emma J Murray http://orcid.org/0000-0003-3345-2698

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
