## [Reviewer comments · BMJ Open]

ARTICLE DETAILS

TITLE (PROVISIONAL)	Sociological Variety and the Transmission Efficiency of Mycobacterium Tuberculosis: A Secondary Analysis of Qualitative and Quantitative Data from Fifteen Communities in Zambia.
AUTHORS	Murray, Emma; Dodd, PJ; Marais, B; Ayles, Helen; Shanaube, Kwame; Schaap, Albertus; White, RG; Bond, Virginia

VERSION 1 – REVIEW

REVIEWER	Koch, Anastasia University of Cape Town, Molecular Mycobacteriology Research Unit, Institute of Infectious Disease and Molecular Medicine
REVIEW RETURNED	28-Feb-2021

GENERAL COMMENTS	OVERALL COMMENTS Thank you for the opportunity to review this manuscript. The data presented in the manuscript comprises a mixed-methods analysis to understand how social factors facilitate the spread of tuberculosis (TB) in 15 neighbourhoods in Zambia. The findings suggest that there are differences in how specific socio-spatial factors influence TB transmission. Social factors were broadly categorised into those that impact spatial crowding and social permeability of neighbourhoods. Importantly, the findings suggest that some of these factors ('routine' as described by the authors) can be addressed via public health and information interventions. In contrast, others ('structural' as defined by the authors) might be more difficult to change over the short-medium term. There are three key strengths of the study: 1. It's critical that the contribution of social factors to the spread of TB is investigated with the same rigour as the biomedical and clinical aspects. While some of the results in the manuscript may seem intuitive (and some have been previously observed in other regions), the fact that interdisciplinary data-driven findings have been presented around how socio-spatial factors impact TB transmission is noteworthy.2. The combination of quantitative (for TB transmission) and qualitative (for socio-spatial factors) is interesting, and as noted by the authors, the interdisciplinarity of the approach is unique.
--

	3. The study uses data from and builds on a previous large-scale interventional study of TB (ZAMSTAR) and thus maximises its findings and usefulness. It's also good to see that the approach to Patient and Public Involvement (PPI) is described in the methods. PPI is not always published in manuscripts, leaving little opportunity to assess if and how it was conducted. Other manuscripts should follow this example, and a description of PPI work in publications should become the accepted standard in the field. While the study's limitations are clearly noted, some additional aspects as described below should be addressed to further strengthen the manuscript. MINOR REVISIONS INTRODUCTION 1. Either here or in the section of the discussion that mentions potential limitations - I think it would be important to note some of the caveats of using the ratio of Annual Risk of Infection (ARI) to TB prevalence to determine transmission efficiency. Others have published valid and relevant caveats that could influence the reliability of the measure, for example, https://bmcinfectdis.biomedcentral.com/articles/10.1186/1471-2334-11-156). Two of the potential caveats that have been previously described: a) that the baseline prevalence of TB is measured in children and b) that the measure can't detect multiple exposures/infections are particularly important in higher-burden settings. While (ARI:TB Prevalence) is a well-known metric and therefore appropriate to use, it's important to mention the caveats. 2. The data the authors have used focus on Zambia. Given that the ZAMSTAR study was conducted in South Africa, it would be useful to know why data for South Africa was not available/included in this study – this could be important for the generalisability of the findings. METHODS 1. It would be useful to describe how neighbourhoods were selected for the study. The original CODA and ZAMSTAR studies are cited here, but it's not clear how many of the same neighbourhoods were included here and a brief description here would also mean the reader would not have to check the methods of previous papers to assess whether there is any potential bias in selection of communities. 2. While observations of 'social permeability' are really important and interesting, there could also be a time component to this – the longer people spend either outside their own neighbourhood or that people from outside of the neighbourhood are present could influence TB transmission risk. Moreover, the nature of activities (e.g., whether in well-ventilated areas
--	--

	or not) could also play a role. Could the authors suggest a way to take these dynamics into account, either here or in the discussion? 3. The potential limitations of the broad-based survey (BBS) and its ability to detect all social and spatial engagements has been well-described. One thing that was noted or mentioned is how household contacts/engagements might interact with non-household engagements to influence TB transmission. Again, this could be mentioned here or in the discussion. 4. Page 9, line 10 -11, could the authors provide a reference for ranking the burden of infectiousness of men's involvement higher than woman's and adults higher than children. These ideas make sense, but a citation would increase validity. RESULTS 1. Could the authors note if some neighbourhoods that were studied more rural or urban, and if this might have contributed to some of the findings. 2. It would be interesting to note if there were differences in the availability of healthcare services between neighbourhoods and whether this could contribute to the overall risk of TB transmission/ 3. Importantly, schools have been noted as potential space for TB transmission: https://journals.plos.org/plosone/article?id=10.1371/journal.pone.0039246. It would be important to mention whether the BBS noted if children travelled outside of neighbourhoods for schooling, and if so, whether this could have influenced TB transmission risk. 4. For people who travelled outside of the neighbourhood or people who travelled into new neighbourhoods, did this always occur via public transport? Could the mode of transportation into or out of neighbourhoods interact with the frequency of travel to impact TB transmission risk? 5. Figure 1 is not clearly annotated in the manuscript and I couldn't see a figure legend. Moreover, given the relatively large amounts of data presented, it might make the figure easier to interpret if some of the data points were colour-coded according to them. For example, structural vs routine socio-spatial factors and/or spatial and permeability factors could be distinguished using different colours. DISCUSSION 1. An essential issue to that could be addressed in more detail in the discussion is the generalisability. The study uncovers interesting and relevant interactions between socio-spatial factors and provides relevant areas for follow up. Moreover, the idea of having contextualised rather than one-size-fits all public health interventions is a good one. However, how would the authors envisage generalising these findings to other regions (where social
--	--

	patterns and co-morbidities are likely to significantly vary)? Given the social uniqueness of different communities, would this kind of study need to be conducted in all high-burden areas to determine neighbourhood-specific interventions? It could be useful to have suggestions for how these kinds of studies could be achieved and contribute to region-appropriate contextualised public health interventions. 2. It could be interesting to discuss how whole-genome sequencing of M. tuberculosis might aid in confirming/ruling out sites of transmission as well as cases that were a result of permeability of the neighbourhoods. 3. Finally, social permeability and levels of spatial crowding likely interact to impact TB transmission risk. It could be interesting to discuss how the data could be analysed, taking these interactions into account.
--	---

REVIEWER	Dara, M World Health Organization Office at the European Union, Brussels, Belgium
REVIEW RETURNED	10-Mar-2021

GENERAL COMMENTS	Congratulations with a great paper. I suggest the following points to improve your paper:  - Please expand on the results of the prevalence survey and the ARI estimates, e.g. which method had been used for the survey and any other consideration relevant to that survey (changes with any previous survey ...). - Please consider adding in the background and in the discussion sessions more data or their absence of any racial differences or other determinants (e.g. nutritional status, level of income prevalence of HIV,) or any other determinants which may have had an impact on the results. Thanks and regards, Dr Masoud Dara
---

VERSION 1 – AUTHOR RESPONSE

Reviewer Comment	Author Response
Reviewer 1 - Introduction	
1. Either here or in the section of the discussion that mentions potential limitations - I think it would be important to note some of the caveats of using the ratio of Annual Risk of Infection (ARI) to TB prevalence to determine transmission efficiency. Others have published valid and	Thanks for these comments. We do agree that ARI measured by regression LTBI prevalence by age etc. may under-estimate infections by not registering re-infection, although this effect is lessened by focusing on children.

relevant caveats that could influence the reliability of the measure, for example, <https://bmcinfectdis.biomedcentral.com/articles/10.1186/1471-2334-11-156>). Two of the potential caveats that have been previously described: a) that the baseline prevalence of TB is measured in children and b) that the measure can't detect multiple exposures/infections are particularly important in higher-burden settings. While (ARI:TB Prevalence) is a well-known metric and therefore appropriate to use, it's important to mention the caveats. Page 2 of 3

Your point that ARI in adults may be different to that measured in children is also well taken, and indeed one of the main arguments in our previous paper (now reference 13).

However, since we are interested in how the relative rather than absolute value of the ARI/prevalence ratio compares across communities, it only matters that the relationship between measured ARI with average infection risks does not vary. Our previous paper [13] does present data showing that reported contact patterns by age (which mediate the relationship between adult and child ARI) show remarkable consistency across communities, which helps defend our use of a metric based on ARI measured in children for our exploratory comparisons.

Additional limitations with this metric include the dependency of ARI on TST interpretation method, and the measurement uncertainty associated with both TB prevalence and ARI estimates.

We have tried to capture some of this by adding to the limitations paragraph in the Discussion as you suggest:

"...in future work. The estimates of ARI and TB prevalence are of limited precision, and ARI estimates can depend strongly on the method of TST interpretation. Empirical estimates [29] and modelling [13] suggest ARI estimates in children are lower than in adults. However, ARI-to-prevalence ratios should still hold validity for relative comparisons between communities, since age-related contact patterns do not vary substantially between these communities. [13]"

We have also clarified the TST method used for the ARI estimates in the quantitative analysis section:

"As part of ZAMSTAR, TST surveys were conducted in a cohort of primary school children to calculate ARI across communities, estimating community ARIs of

	between 0.25%/year and 1.49%/year (using a 15mm cut off for positivity). [16]
2. The data the authors have used focus on Zambia. Given that the ZAMSTAR study was conducted in South Africa, it would be useful to know why data for South Africa was not available/included in this study – this could be important for the generalisability of the findings.	Our analysis was heavily reliant on BBS Data from the ancillary study, CODA. The qualitative component of CODA was conducted in the Zambian ZAMSTAR communities only, primarily due to practical and financial constraints. Therefore, there is no equivalent set of BBS data available from the South African sites. We have made the following minor adjustment to the Introduction to flag that CODA was integral and only conducted in Zambia: “We used a novel mixed methods interdisciplinary approach to conduct a secondary analysis of Zambian data sets, collected during a cluster randomised trial (CRT) - the Zambia and South Africa TB and AIDS Reduction Study (ZAMSTAR) - and as part of an ancillary study, named the Contact Observations of Daily Activities (CODA). [12][13]” Owing to the relational and qualitative foundation of the analysis, findings are necessarily context specific. Having said this, the BBS method itself, which allows researchers to develop contextually informed indicators, is easily transferable and the dual dynamic of social permeability and crowding could be relevant to variations in transmission efficiency across other settings. To convey this, we have adapted the final Discussion paragraph on limitations (new text underlined): “Findings are context specific and data were effectively cross-sectional. We focused on interactions within localised geographical boundaries, and did not capture interactions on the far side of extra-community travel. Furthermore, fieldwork was conducted at a particular time of year and observations took place during the daytime, which limits the generalisability of our observations accordingly – not only across but also within the study communities. We anticipate relevant socio-spatial features and their role in mediating TB transmission will vary between contexts. However, the dual dynamic of spatial closeness (crowding) and social

	openness (permeability) may be relevant in other settings, with further investigations valuable. In addition, we revised our Summary of strengths and limitations to include: “Results are context specific, therefore further investigation in other settings would be valuable to assess the generalisability of key concepts.”
Reviewer 1 - Methods	
1. It would be useful to describe how neighbourhoods were selected for the study. The original CODA and ZAMSTAR studies are cited here, but it's not clear how many of the same neighbourhoods were included here and a brief description here would also mean the reader would not have to check the methods of previous papers to assess whether there is any potential bias in selection of communities.	Thank you, yes, something similar is highlighted by Reviewer 2. We have reformatted our abstract to flag important parameters of the setting and included the following text at the end of the introductory paragraph to Methods: “The Zambian communities we investigated in this secondary analysis were pre-selected as a set of sixteen for the ZAMSTAR trial. The ZAMSTAR research design needed the potential to detect statistically significant reductions in TB prevalence and infection incidence. Therefore, communities were selected based on pre-specified criteria including TB case notification > 400/100,000. A further prerequisite was that TB diagnostic and treatment service were broadly equivalent (de facto functioning local health facilities provided by the state, with trained health care providers and voluntary TB supporters, serving a population of at least 25 000). Given the role of HIV infection in driving TB cases in this setting, ZAMSTAR additionally selected on the basis of relatively high burdens of HIV (compared to the national average). In the absence of population level data on HIV, proxy information was used to identify communities, drawing on the opinion of local experts, other survey sources and seeking the buy-in and advice from the Ministry of Health. The resulting Zambian ZAMSTAR communities were distributed across five provinces and districts with subsequent measures by the ZAMSTAR trial showing that HIV prevalence ranged between 8.1 and 26.6%. [12] All communities were predominantly high density urban communities (three were peri-urban) located along the line of rail and/or major trading routes. All communities had a broad mix of ethnic groups and four communities had a stronger, yet insignificant presence of non-nationals due to the proximity of international borders and location within the capital city. Socio-economic status was mainly classified

	as low, with some pockets of lower middle-class in all communities.” Also see minor change at end of ‘qualitative data’ section in Methods (new text underlined here): “One of the sixteen Zambian ZAMSTAR communities was dropped from our analysis due to poor data quality. Similarities and differences were subsequently compared across the remaining fifteen communities.”
2. While observations of ‘social permeability’ are really important and interesting, there could also be a time component to this – the longer people spend either outside their own neighbourhood or that people from outside of the neighbourhood are present could influence TB transmission risk. Moreover, the nature of activities (e.g., whether in well-ventilated areas or not) could also play a role. Could the authors suggest a way to take these dynamics into account, either here or in the discussion?	Yes, both the time element and the nature of activities is definitely important and highly complex. While the BBS didn’t include quantitative measures, data do provide a sense of both these dimensions of socio-space and this did help guide us with the collapsing of data into meaningful indicators and variables, significant to the life of communities. So, in a qualitative way, these dynamics are captured throughout the analysis, remembering that whole communities, rather than individuals, were the unit of analysis. Unfortunately, we don’t have quantitative measures from BBS of time spent outside of residential neighbourhoods or of how long people from other locations of the community were present. This is a limitation of the BBS as it has been applied to date, yet, recent research is showing that quantitative measures of geographic mobility could easily be applied in future studies. Please see how new reference below [37] has been added to the revision of the final paragraph of Discussion: Robsky KO, Isooba D, Nakasolya O, et al. Characterization of geographic mobility among participants in facility- and community-based tuberculosis case finding in urban Uganda. PLoS One. 2021;16(5):e0251806. Published 2021 May 14. doi:10.1371/journal.pone.0251806 Furthermore, BBS does not capture extra-community interactions. In other words, our data were focused on what was happening within residential communities and

	not on what happened in places outside of the communities, where local residents frequently spent time. We have reworked our Summary section of limitations and strengths to emphasise this early on in paper. New text in Summary: “Data were effectively cross-sectional, focusing on interactions within localised geographical boundaries, and did not capture details of extra-community travel, nor of infrequent and/or seasonal events, potentially relevant to TB transmission.” Also see minor addition to beginning of methods: “This secondary analysis used whole communities, rather than the individual, as a unit for analysis and...”
3. The potential limitations of the broad-based survey (BBS) and its ability to detect all social and spatial engagements has been well-described. One thing that was noted or mentioned is how household contacts/engagements might interact with non-household engagements to influence TB transmission. Again, this could be mentioned here or in the discussion.	We assume that this comment should have read “one thing that was not noted or mentioned is...”, and we have responded accordingly. The BBS method used to generate qualitative data is rapid and focused on communities at large, providing a broad-brush picture of the collective. It is a valid method for participant observation in public spaces. Less so within households, although day long household observations were conducted to understand who was staying at home and where residents went when they left. We argue that, for a valid qualitative understanding of household contacts and how they interact with the broader community, one would need deeper ethnographic work, directed at a selection of households. We opted for the community level focus due to transmission being more likely to take place in the wider community in this particular setting, although targeting tuberculosis-affected households for tuberculosis screening, HIV testing, and referral for

	treatment of tuberculosis or M. tuberculosis infection remains a priority. While households are not the focus of this particular study, they are obviously part of the sociological topography of communities. BBS does capture this, both through the 'Housing Density' indicator and the indicator, 'Who Remains at Home'. Interestingly, households did not emerge in this BBS data as significant 'hotspots' for community transmission. We are aware of other settings where households have emerged as hotspots. This has been in cases where boundaries between the private home and wider community are particularly blurred, e.g. where private households start functioning more as public churches. In addition, we do include where children are coming into contact with adults in the broader community and this does start to give an indication of where household contacts are engaging with others (e.g. video shows, bars and on transport). In sum, we feel that the design of our qualitative methods and the study setting, mean that the delineation of gender and age is more appropriate than that between household and community. We addressed this point by adding the following text with new reference to our 'Qualitative Data' segment in Methods, rather than the discussion, as it pertains more to the framework of our study: "The qualitative BBS data indicated how public places and transport were used by different age groups and genders within each community. Our focus on the broader community is justified as we know that transmission is less likely to take place within households in this particular setting. [21]"
4. Page 9, line 10 -11, could the authors provide a reference for ranking the burden of infectiousness of men's involvement higher than woman's and adults higher than children. These ideas make sense, but a citation would increase validity.	Thanks for this. We have modified this sentence ('in Mixed Methods Analysis'). Old version: "Expectations of the burden of infectious tuberculosis led to men's involvement being ranked higher than

	women's involvement, and adults' involvement higher than children's." New version: "Previous work on these communities combining the TB prevalence results, ARI estimates and quantitative social contact data, suggested men are responsible for more infections than women. [13] Children, especially age <5 years, typically have lower bacteriological load than adults and are likely to contribute less to transmission than adults of either sex."
Reviewer 1 - Results	
1. Could the authors note if some neighbourhoods that were studied more rural or urban, and if this might have contributed to some of the findings.	The setting is mostly urban and there is no doubt that this contributed to our findings: BBS is a participatory and the observational qualitative method is by definition contextually sensitive. Three of the sixteen Zambian ZAMSTAR communities were classed as peri-urban with an urban centre (in a district town or on a major road), with certain parts spreading into more rural areas. We did not conduct BBS in the rural spread of these communities, so our findings are from urban and peri-urban settings. We have restructured our abstract, also in line with the Editor's comments, to add the following: "Setting: ZAMSTAR communities in urban and peri-urban Zambia, spanning five provinces." Please also see new text in Methods, comment 1 (supra), which includes the following: "...All communities were predominantly high density urban communities (three were peri-urban) located along the line of rail and/or major trading routes. All communities had a broad mix of ethnic groups and four communities had a stronger, yet insignificant presence of non-nationals due to the proximity of international borders and location within the capital city."
2. It would be interesting to note if there were differences in the availability of healthcare services	This is an important consideration - thank you for querying. Functioning state TB diagnostic centres were

between neighbourhoods and whether this could contribute to the overall risk of TB transmission/	located within each community as a prerequisite for the parent study, ZAMSTAR. We have revised our abstract to highlight this. Please see: “Participants: Fifteen communities, each served by a health facility offering tuberculosis treatment to a population of at least 25,000. TB notification rates were at least 400 per 100,000 per annum and estimates of HIV seroprevalence high.” Healthcare services were therefore comparable to the extent that they all provided TB diagnosis and treatment; had trained health care providers; and voluntary TB supporters. A degree of parity is implied. The maximum distances needed to travel to the diagnostic centre did vary across communities, but it was beyond the scope of this piece to assess the impact of any nuances in state healthcare services on the overall risk of TB transmission, beyond ‘crowding’ at facilities, as captured by the indicator “Crowding at Health Centres” (Table 1) Please also see new text in Methods, comment 1 (supra). Which includes the following text: ...”A further prerequisite was that TB diagnostic and treatment service were broadly equivalent (de facto functioning local health facilities provided by the state, with trained health care providers and voluntary TB supporters, serving a population of at least 25,000).”
3. Importantly, schools have been noted as potential space for TB transmission: https://journals.plos.org/plosone/article?id=10.1371/journal.pone.0039246. It would be important to mention whether the BBS noted if children travelled outside of neighbourhoods for schooling, and if so, whether this could have influenced TB transmission risk.	Schools do play an important role within the communities and formed sites for TST testing for the ZAMSTAR trial. Therefore, in an indirect way, transmission in schools is accounted for in the ARI data: Infection in school children was used as marker of what was going on in the broader community. This secondary analysis, however, was not investigating potential hotspots for transmission, but rather trying to classify, or type, sociological variation in meaningful way for transmission efficiency. BBS findings did show that in four communities, a notable number of children were observed out of school.

	Please see Table 1, 4th to last row. We couldn't identify whether these children were not attending school, or simply had a different time structure to their school day and were unable to meaningfully investigate. There was one community where children were observed travelling to another community to attend secondary school, but again this didn't emerge as significant. As mentioned earlier, a limitation of the BBS is that it doesn't document extra-community activity. Our text has been revised to include this. Please see revised Summary text and Discussion, as highlighted in comment 2 for Methods (supra) and adjusted text to limitations paragraph in Discussion, with new reference: Old version: “This analysis does not report on all significant features linked to social and spatial engagement in the communities. We do not pay much attention to the large amount of time residents spent at church and no substantial differences presented across communities, yet congregation of this nature has been established as a transmission risk factor in high burden settings [27].” New version: “Furthermore, we emphasise that the analysis did not include all community features relevant to social-spatial engagement. Congregation linked to both schools and churches has been established as a transmission risk factor in similar high burden settings [30, 31], yet we were unable to conduct BBS at schools. Importantly, churches emerged as significant (and mostly crowded) congregate features, with widespread attendance presenting in similar ways across age, gender and time. This consistency, across all communities, meant that churches could not be linked to sociological variation.”
4. For people who travelled outside of the neighbourhood or people who travelled into new neighbourhoods, did this always occur via public transport? Could the mode of transportation into or out of neighbourhoods interact with the frequency of travel to impact TB transmission risk?	This is an interesting question. However, in almost all instances minibuses were the predominant mode of extra-community travel, except in one community where residents predominantly travelled by foot. Please see page 11, lines 6-15, of original submission. As such, we are not able to meaningfully investigate this.

5. Figure 1 is not clearly annotated in the manuscript and I couldn't see a figure legend. Moreover, given the relatively large amounts of data presented, it might make the figure easier to interpret if some of the data points were colour-coded according to them. For example, structural vs routine socio-spatial factors and/or spatial and permeability factors could be distinguished using different colours.	Thanks for this suggestion. We have improved the annotation of Figure 1 (and similar supplementary files) by using colour distinctions between the indicators of crowding, indoor activity and social permeability. In fact, we did provide legends/captions for the figures, but these did not appear in the reviewer version alongside the figures. The captions are: Figure 1: Transmission Efficiency (ratio of Annual Risk of Infection to Disease Prevalence) by Socio-Spatial Variables Figure S1 Annual Risk of Infection by Socio-Spatial Variables (supplemental material) Figure S2: Prevalence of Disease by Socio-Spatial Variables (supplemental material)
Reviewer One - Discussion	
1. An essential issue that could be addressed in more detail in the discussion is the generalisability. The study uncovers interesting and relevant interactions between socio- spatial factors and provides relevant areas for follow up. Moreover, the idea of having contextualised rather than one-size-fits all public health interventions is a good one. However, how would the authors envisage generalising these findings to other regions (where social patterns and co-morbidities are likely to significantly vary)? Given the social uniqueness of different communities, would this kind of study need to be conducted in all high-burden areas to determine neighbourhood-specific interventions? It could be useful	Owing to the exploratory and hypothesis generating nature of the work, generalisability is limited. Also see related response to Introduction, pt 2 (supra). Theoretically, the BBS method is designed to be implemented in more than one neighbourhood/community where there are tangible and cogent socio-geographical boundaries and where similar features can be compared. Practically, a particular research interest and logistical limitations will shape the number of communities researched and for what purpose BBS can take place. It is a method that lends itself to systematic comparison of similarities and differences across communities and countries and thereby, using a framework of meta-indicators, lends itself to generalisability around a core research question. We would therefore argue that the pattern of socio-spatial factors could be relevant to other urban places and BBS would be an appropriate rapid method to assess key features for the purposes of comparison.

to have suggestions for how these kinds of studies could be achieved and contribute to region-appropriate contextualised public health interventions.	We have added the following text with new references to the 'Qualitative Data' section of Methods to help explain how the method is applied: “...communities as part of CODA.[17] BBS is a method that lends itself to the systematic sociological comparison of bounded urban places (in this instance communities) across socio-space. The development of meta-indicators around a core research question allows for abstraction and a degree of generalisability. [17-20] The CODA BBS aimed to..” Please also note included references relevant to research from similar sub-Saharan settings, showing that some findings might be generalizable. See References [24-27].
2. It could be interesting to discuss how whole-genome sequencing of M. tuberculosis might aid in confirming/ruling out sites of transmission as well as cases that were a result of permeability of the neighbourhoods.	This is an exciting and clear area for future interdisciplinary research. BBS can be used in many different settings for both the qualitative identification of local transmission 'hotspots' and the classification of communities across a grading of social permeability. Whole genome sequencing could help clarify whether importation is more or less relevant in certain communities and highlight transmission chains. We have added the following to the end of Discussion: “...The addition of pathogen genomic data might help generate or eliminate hypotheses on transmission location or context, and recent research is showing how quantitative measures of geographic mobility could easily be applied to studies of social permeability. [37] Whole genome sequencing could clarify whether importation is more or less relevant in certain communities and highlight transmission chains.” And the following at the end of Conclusion: “The significance of social permeability could be further explored by integrating community level typologies with

	genome sequencing techniques able to map strain introduction event and community transmission chains.”
3. Finally, social permeability and levels of spatial crowding likely interact to impact TB transmission risk. It could be interesting to discuss how the data could be analysed, taking these interactions into account	For a more in-depth analysis of how social permeability and spatial crowding interact, we would advocate on a more limited scope, focusing on one community, which, for us, would require more data collection. The best way that we could come up with in our study, given the breadth across communities and nature of the data, was the ability to follow a particular community (number) through the graphics of FIGURE 1, which provides a sense, albeit compact and complex, of how permeability and crowding interact within a particular community. For example, if one takes community 14 into consideration, we see that although non-residents are coming into this community, crowding in public space is less than in most other communities and transport is not overloaded and poorly ventilated. For community 4, with higher transmission efficiency, non-residents are also coming in, but public spaces and transport were gauged as more crowded. We have adjusted ‘Patterns with respect to transmission efficiency’ in Results to highlight how Figure 1 should be used in this way, in order to analyse interaction: “By tracing an individual community (number) through Figure 1, one is able to consider the interaction of the different socio-spatial variables identified. Where transmission efficiency was highest...” We think our changes above and minor tweaks in paragraph 2 of the Discussion create more clarity on how our qualitative work embraces, rather than reduces, this complexity. We hope this is sufficiently improved.
Reviewer Two – General Comments	
1. Please expand on the results of the prevalence survey and the ARI estimates, e.g. which method had been used for the survey and any other consideration relevant to that survey (changes with any previous survey ...)	Thanks for this comment. As we noted, the methods for the historic TB prevalence and TST surveys whose data we are using have been described in references [14-16]. However, we have added some relevant methodological and background results to save readers referring to these papers.

	In the Introduction, we have edited the sentence describing the prevalence survey results to: “The final ZAMSTAR prevalence survey found large variation in culture-confirmed TB prevalence, from 221 to 1095 per 100,000 in the Zambian communities.” We have added to the sentence in the Quant Methods section to give a sense of the range of ARIs estimated, and also to now state the methods used in interpreting the TST results on which these ARI results are based. This sentence now reads: “As part of ZAMSTAR, TST surveys were conducted in a cohort of primary school children to calculate ARI across communities, estimating community ARIs of between 0.25%/year and 1.49%/year (using a 15mm cut off for positivity). [16]”
2. Please consider adding in the background and in the discussion sections more data or their absence of any racial differences or other determinants (e.g. nutritional status, level of income prevalence of HIV,) or any other determinants which may have had an impact on the results.	Thank you for requesting. Reviewer 1, in a similar vein, also felt more explanation of selection criteria and study setting was needed. We have reformatted our Abstract to flag important parameters of the setting: “Setting: ZAMSTAR communities in urban and peri-urban Zambia, spanning five provinces. “Participants: Fifteen communities, each served by a health facility offering tuberculosis treatment to a population of at least 25,000. TB notification rates were at least 400 per 100,000 per annum and estimates of HIV seroprevalence high. We have also included the following text at the end of the introductory paragraph to the Methods section: “The Zambian communities we investigated in this secondary analysis were pre-selected as a set of sixteen for the ZAMSTAR trial. The ZAMSTAR research design needed the potential to detect statistically significant reductions in TB prevalence and infection incidence. Therefore, communities were selected based

	on pre-specified criteria including TB case notification > 400/100,000. A further prerequisite was that TB diagnostic and treatment service were broadly equivalent (de facto functioning local health facilities provided by the state, with trained health care providers and voluntary TB supporters, serving a population of at least 25 000). Given the role of HIV infection in driving TB cases in this setting, ZAMSTAR additionally selected on the basis of relatively high burdens of HIV (compared to the national average). In the absence of population level data on HIV, proxy information was used to identify communities, drawing on the opinion of local experts, other survey sources and seeking the buy-in and advice from the Ministry of Health. The resulting Zambian ZAMSTAR communities were distributed across five provinces and districts with subsequent measures by the ZAMSTAR trial showing that HIV prevalence ranged between 8.1 and 26.6%. [12] All communities were predominantly high density urban communities (three were peri-urban) located along the line of rail and/or major trading routes. All communities had a broad mix of ethnic groups and four communities had a stronger, yet insignificant presence of non-nationals due to the proximity of international borders and location within the capital city. Socio-economic status was mainly classified as low, with some pockets of lower middle-class in all communities.” Please also see substantial revisions to the last two paragraphs of our Discussion, covering strengths and limitations.
--	--

VERSION 2 – REVIEW

REVIEWER	Koch, Anastasia University of Cape Town, Molecular Mycobacteriology Research Unit, Institute of Infectious Disease and Molecular Medicine
REVIEW RETURNED	19-Jul-2021
GENERAL COMMENTS	Thank you for the thorough addressing of all comments. I have no further comments and recommend publication.